# Loganin Attenuates High Glucose-Induced Schwann Cells Pyroptosis by Inhibiting ROS Generation and NLRP3 Inflammasome Activation

**DOI:** 10.3390/cells9091948

**Published:** 2020-08-23

**Authors:** Yu-Chi Cheng, Li-Wen Chu, Jun-Yih Chen, Su-Ling Hsieh, Yu-Chin Chang, Zen-Kong Dai, Bin-Nan Wu

**Affiliations:** 1Department of Pharmacology, Graduate Institute of Medicine, College of Medicine, Drug Development and Value Creation Research Center, Kaohsiung Medical University, Kaohsiung 80708, Taiwan; u9251055@gmail.com (Y.-C.C.); cheni9420@gmail.com (Y.-C.C.); 2Department of Nursing, and Department of Cosmetic Application and Management, Yuh-Ing Junior College of Health Care and Management, Kaohsiung 80776, Taiwan; juliechu0101@gmail.com; 3Division of Neurosurgery, Fooyin University Hospital, Pingtung 92847, Taiwan; drchen0724@yahoo.com.tw; 4School of Nursing, Fooyin University, Kaohsiung 83102, Taiwan; 5Department of Pharmacy, Kaohsiung Medical University Hospital, Kaohsiung 80708, Taiwan; soolinchia@yahoo.com.tw; 6Department of Pediatrics, School of Medicine, College of Medicine, Kaohsiung Medical University, Kaohsiung 80708, Taiwan; 7Department of Pediatrics, Division of Pediatric Cardiology and Pulmonology, Kaohsiung Medical University Hospital, Kaohsiung 80708, Taiwan; 8Department of Medical Research, Kaohsiung Medical University Hospital, Kaohsiung 80708, Taiwan

**Keywords:** Schwann cell, loganin, high glucose, reactive oxygen species, NLRP3 inflammasome, pyroptosis

## Abstract

Diabetic peripheral neuropathy (DPN) is caused by hyperglycemia, which induces oxidative stress and inflammatory responses that damage nerve tissue. Excessive generation of reactive oxygen species (ROS) and NOD-like receptor protein 3 (NLRP3) inflammasome activation trigger the inflammation and pyroptosis in diabetes. Schwann cell dysfunction further promotes DPN progression. Loganin has been shown to have antioxidant and anti-inflammatory neuroprotective activities. This study evaluated the neuroprotective effect of loganin on high-glucose (25 mM)-induced rat Schwann cell line RSC96 injury, a recognized in vitro cell model of DPN. RSC96 cells were pretreated with loganin (0.1, 1, 10, 25, 50 μM) before exposure to high glucose. Loganin’s effects were examined by CCK-8 assay, ROS assay, cell death assay, immunofluorescence staining, quantitative RT–PCR and western blot. High-glucose-treated RSC96 cells sustained cell viability loss, ROS generation, NF-κB nuclear translocation, P2 × 7 purinergic receptor and TXNIP (thioredoxin-interacting protein) expression, NLRP3 inflammasome (NLRP3, ASC, caspase-1) activation, IL-1β and IL-18 maturation and gasdermin D cleavage. Those effects were reduced by loganin pretreatment. In conclusion, we found that loganin’s antioxidant effects prevent RSC96 Schwann cell pyroptosis by inhibiting ROS generation and suppressing NLRP3 inflammasome activation.

## 1. Introduction

Diabetic neuropathy is a common microvascular complication of both type 1 and type 2 diabetes mellitus. The global prevalence of diabetic peripheral neuropathy (DPN) is ~30% in type 2 diabetic patients and ~26% in type 1 diabetic patients with an overall incidence of ~29% [1,2]. It is generally agreed that hyperglycemia mainly contributes to the development of diabetic neuropathy or DPN. DPN caused neuroinflammation, peripheral nerve injury, reduced nerve conduction velocity and altered the nerve fiber Na^+^, K^+^-ATPase activity, which provokes reactive oxygen species (ROS) release [3]. DPN is the most frequently occurring complication in type 2 diabetes and often affects sensory and motor neurons of peripheral nerves causing pain and discomfort in lower extremities, risk of foot ulcerations and reduced quality of life [4,5]. Hyperglycemia induces the overproduction of free radicals, especially ROS, recognized as a dominant factor in DPN pathogenesis [6].

They are the most numerous glial cells of peripheral nerves and maintain peripheral nerve structure and function by ensheathment of unmyelinated axons, myelination of myelinated axons and secretion of neurotrophic factors [7,8,9]. Schwann cells are very sensitive to glucose and insulin levels and closely involved in DPN pathogenesis. Numerous reports have demonstrated that hyperglycemia causes Schwann cell apoptosis during the development of DPN [10,11,12,13]. Schwann cells play a crucial role in DPN pathogenic mechanisms and are thus an important therapeutic target for diabetic neuropathy [7,8,9].

Inflammasomes are an innate immune response in aging and diabetes and other neurodegenerative diseases and are a new therapeutic target for diabetic complications [14,15]. Some chronic inflammatory conditions, such as endometriosis, bronchiolitis obliterans syndrome, etc. have been shown to regulate NLRP3 inflammasome via the Formyl Peptide Receptor 1 (Fpr-1)-mediated signaling pathway [16,17]. Many types of inflammasomes exist in various cell types. NLRP3 (NOD-like receptor protein 3) inflammasomes are the most widely studied [15,18]. The NLRP3 inflammasome is composed of the sensor molecule NLRP3, the adaptor protein ASC (apoptosis-associated speck-like protein containing a caspase recruitment domain) and pro-caspase-1. The activation of NLRP3 inflammasomes is strictly regulated to avoid excessive inflammatory responses [17,19]. NLRP3 inflammasome activation requires a two-step process, a priming step and an activation step. During the priming step, endogenous cytokines or pattern recognition receptors (PRRs) provoke nuclear factor-κB (NF-κB) activation. NF-κB then promotes the transcription of NLRP3, pro-Interleukin (IL)-1β and pro-IL-18. The activation step is the inflammasome assembly, triggered by several molecular and cellular events including K^+^ efflux, Ca^2+^ signaling and ROS generation. During the assembly of NLRP3 inflammasomes, pro-caspase-1 transitions to cleaved-caspase-1, which promotes the cleavage of pro-IL-1β and pro-IL-18 precursors to mature forms. In this process, activated caspase-1 also cleaves gasdermin D (GSDMD) into the N-terminal of GSDMD (GSDMD–NT), forming pores in the plasma membrane and then induces inflammatory programmed cell death, called pyroptosis [20,21,22].

Loganin is an iridoid glycoside isolated from the fruit *Cornus officinalis*. Many studies have confirmed that loganin, traditionally used to treat diabetic nephropathy, has antioxidant, anti-inflammatory and hypoglycemic effects [23,24,25]. By inhibiting the advanced glycation end-products (AGEs) pathway and inhibiting aldose reductase, loganin can alleviate liver and kidney damage and other diabetic complications related to metabolic abnormalities caused by oxidative stress, inflammation and apoptosis [26,27,28,29]. Loganin improves inflammation in acute pancreatitis and lung diseases by inhibiting the activation of NF-κB [30]. Loganin possibly also ameliorates depression- and anxiety-like behaviors associated with diabetes by lowering blood glucose and proinflammatory cytokine levels [31]. In terms of neuroprotection, loganin also attenuates mesencephalic neuronal apoptosis, neurite damage and oxidative stress through enhancement of neurotrophic factors [32]. Furthermore, loganin has been demonstrated to mitigate neuropathic pain by preventing Schwann cell demyelination in rats with chronic contraction injury [33].

To date, little is known about the effect of loganin on high-glucose-induced NLRP3 inflammasome activation and the downstream cascades resulting in pyroptosis. This study examined the impact of high glucose on Schwann cells and whether loganin could protect against high-glucose-induced cell death.

## 2. Materials and Methods

### 2.1. Cell Culture and Drug Treatment

RSC96 cells (rat Schwann cell line, BCRC No. 60507) were purchased from Bioresource Collection and Research Center, Food Industry Research and Development Institute (Hsinchu, Taiwan). RSC96 cells were maintained in Dulbecco’s modified Eagle’s medium (DMEM) (Thermo Fisher Scientific, Waltham, MA, USA) containing 5.6-mM glucose, 4-mM L-glutamine plus with 100 U/mL penicillin, 100-µg/mL streptomycin and 10% (v/v) fetal bovine serum (Thermo Fisher Scientific) at 37 °C in 5% CO_2_ humidified atmosphere. The medium was changed every 2–3 days.

At 70% confluence cells were synchronized by serum starvation for 4 hr. RSC cells were pretreated with various concentrations (0.1, 1, 10, 25, 50 μM) of loganin (#19997, Cayman Chemical, Ann Arbor, MI, USA) or 1-mM N-acetyl-L-cysteine (NAC, A9165, Sigma-Aldrich, St. Louis, MO, USA) for 2 h and then treated with normal glucose (NG; 5.6-mM glucose), NG plus mannitol as an osmotic control (5.6-mM glucose + 19.4-mM mannitol) and high glucose (HG; 25-mM glucose) for 24, 48 and 72 h. All experiments were performed within 10 cell passages.

### 2.2. Cell Viability Assays

Cell viability was detected with a Cell Counting Kit-8 (CCK-8, Biotools, Taipei, Taiwan). RSC96 cells were seeded into 96-well plates at a density of 2 × 10^3^ cells/well and allowed to adhere overnight. Then cells were cultured in medium with 25-mM-high glucose and 0.1, 1, 10, 25 and 50-μM loganin for 24, 48 and 72 h. After treatment, the medium was refreshed and 10 μL of the CCK-8 solution was added to each well. After incubation for 2 h at 37 °C at 5% CO_2_, optical absorbance at 450 nm (with 650 nm as reference) was measured using a microplate reader (Synergy™ H1, BioTek, Winooski, VT, USA). An average was calculated from three independent experiments. Cell viability was calculated as [(Treated: A_450_–A_650_) − (Blank:A_450_–A_650_)]/[(control:A_450_–A_650_) − (Blank:A_450_–A_650_)] × 100%.

### 2.3. Measurement of Intracellular Reactive Oxygen Species

Intracellular ROS levels were detected using a 2′,7′-dichlorofluorescin-diacetate (DCFH_2_–DA, D6883, Sigma-Aldrich, St. Louis, MO, USA) dye method. In brief, cells were seeded on 96-well plates with 2 × 10^3^ cells per well and incubated with 25-mM glucose at 37 °C for 4, 24, 48 and 72 h with or without loganin. After the treatments, we added HBSS-washed cells with 10-μM DCFH_2_–DA in a new culture medium at 37 °C for 30 min. The ROS-associated fluorescence intensity was determined with a fluorescence spectrophotometer (Synergy™ H1, BioTek) using excitation and emission wavelengths at 485 and 528 nm, respectively. From the DCF fluorescence, we measured intracellular ROS with a Zeiss LSM710 (Carl Zeiss, Oberkochen, Germany) equipped with Zen software to process the image and analyzed excitation (480 nm) and emission (525 nm) wavelengths by flow cytometry (Attune NxT flow cytometer; Thermo Fisher Scientific, Waltham, MA, USA).

### 2.4. ELISA Measurement of IL-1β Release

IL-1β concentrations were measured using a Rat ELISA kit (RK00009, ABclonal, Woburn, MA, USA). The culture supernatant of RSC96 cells (50 µL) and standards were pipetted in triplicate into appropriate microtiter wells and the assay was performed according to the ELISA kit’s instructions. The absorbance was measured at 450 nm with wavelength correction set at 650 nm (Synergy™ H1, BioTek, Winooski, VT, USA).

### 2.5. Western Blot Analysis

RSC96 cells were seeded in 10-cm dish (2 × 10^5^ cells) and treated with 25-mM HG, 1-μM loganin or both. Then, cells were lysed in M-PER (mammalian protein extraction reagent, 78501, Thermo Fisher Scientific, Waltham, MA, USA) containing EDTA-free protease inhibitor cocktail and PhosSTOP phosphatase inhibitor (Roche Diagnostics, Mannheim, Germany). The lysates were separated by SDS-PAGE (7.5–12%) transferred to PVDF membrane in blocking buffer (TBST containing 3% bovine serum albumin) and incubated with primary antibodies at 4 °C overnight. Purified mouse anti-rat β-actin (A5441, Sigma-Aldrich, St. Louis, MO, USA) was used to normalize the signals generated from anti-phospho-NF-κB (#3033, Cell Signaling, Danvers, MA, USA), NF-κB (#6956, Cell Signaling), NLRP3 (ab214185, Abcam, Cambridge, UK,), ASC (SAB4501315, Sigma-Aldrich), caspase-1 (A0964, ABclonal, Woburn, MA, USA), IL-1β (ab9722, Abcam), IL-18 (ab191860, Abcam), GSDMD (#93709, Cell Signaling) and TXNIP (#14715, Cell Signaling). The membrane was washed with 1× TBS-T (TBS containing 0.1% Tween-20) and the bound antibodies were visualized by developing with ECL western blotting detection reagents (#WBKLS0500, Millipore, Temecula, CA, USA).

### 2.6. Cell Death Assay

Cell death was assessed by staining with Hoechst 33342 (B-2261, Sigma-Aldrich, St. Louis, MO, USA) and propidium iodide (PI, P4170, Sigma-Aldrich) followed by fluorescence microscopy (Zeiss LSM710; Carl Zeiss, Oberkochen, Germany). Briefly, RSC96 cells were plated at an initial density of 1 × 10^4^ cells/well in 24-well plates. After treatments, the nuclei were stained by Hoechst 33342 (10 μM) for 15 min at room temperature, and then dead cells were stained with PI (5 μg/mL) for 15 min in the dark. The cells were observed immediately by Zeiss LSM710 microscope, capturing six random fields for each group. Cell death was quantitated as the percentage of PI-positive cells relative to the total cell number (Hoechst 33342-positive cells). All experiments were performed at least three times.

### 2.7. Immunocytochemistry

RSC96 cells cultured on glass coverslips were fixed with 10% neutral buffered formalin and permeabilized with 0.1% Triton X-100 in phosphate buffered saline (PBS). After blocking with 3% bovine serum albumin in PBS, the coverslips were incubated with primary antibodies to phospho-NF-κB (#3033, Cell Signaling, Danvers, MA, USA) and GSDMD (A10164, ABclonal, Woburn, MA, USA) overnight at 4 °C. Next, the coverslips were incubated with Alexa Fluor 488 goat anti-rabbit IgG for 1 h at room temperature. For double-labeled immunofluorescence, cells were incubated with NLRP3 (ab214185, Abcam, Cambridge, UK,) overnight at 4 °C, followed by Alexa Fluor 488 goat anti-rabbit IgG and then incubated with ASC (SAB4501315, Sigma-Aldrich, St. Louis, MO, USA) and Alexa Fluor 594 donkey anti-rabbit IgG. Alexa Fluor 488 and 594 are green and red fluorescent dyes, respectively. Coverslips were mounted with Fluoroshield™ with DAPI (F6057, Sigma-Aldrich, blue fluorescent dye) and images were acquired by a Zeiss LSM710 equipped with Zen software to process the image.

### 2.8. RNA Extraction, cDNA Synthesis and Quantitative Real-Time PCR (qRT-PCR)

Total RNA was isolated using the PureLink^®^ RNA Mini Kit (Invitrogen, Carlsbad, CA). Briefly, 1 μg RNA was used to synthesize cDNA using the high capacity cDNA reverse transcription kit (Applied Biosystems, Carlsbad, CA, USA). qRT-PCR was performed using TOOLS 2X SYBR RT-qPCR Mix (FPT-BB05, Biotools, Taipei, Taiwan) on the StepOnePlus Real-time PCR system (Applied Biosystems). PCR reaction was carried out with the following temperature profile: 95 °C for 10 min, followed by 40 cycles of 95 °C for 15 sec and 60 °C for 1 min. Relative expression levels of the mRNA were normalized to β-actin. The gene expression was calculated by the 2^−ΔΔCt^. The primer sequences used for qPCR assay are listed in Table 1.

### 2.9. Statistical Analysis

All data were expressed as mean ± standard error (SE) of triplicate experiments. Differences between groups were analyzed using one-way analysis of variance (ANOVA) followed by *Tukey–Kramer* post hoc test. Statistical differences were set at *p* < 0.05 and indicated by asterisks in figures.

## 3. Results

### 3.1. Loganin Effects on Cell Viability in High-Glucose-Treated RSC96 Schwann Cells

The American Diabetes Association defined an average fasting plasma glucose level < 5.6 mM; severe hyperglycemia reaches the glucose level > 22.2–25-mM [34]. To simulate an uncontrolled diabetic state, we designed to culture the cells in 25-mM glucose and investigated the effect of high glucose on the viability of RSC96 cells. The 5.6-mM glucose medium is close to physiological levels [34,35,36,37]. Cell viability was measured by CCK 8 (cell counting kit 8) assay. RSC96 cells were cultured with 25-mM HG for 24, 48 and 72 h. To exclude the osmotic effects caused by 25-mM HG, thus, 5.6-mM NG with 19.4-mM mannitol was incubated for 72 h and used as an osmotic control. After 25-mM HG incubation, RSC96 cell viability decreased at 48 and 72 h than 5.6-mM NG, but no significant effects were found at 24 h. There were no significant differences between NG with mannitol and NG groups found, and therefore the osmotic effects could be excluded (Figure 1A). Loganin at the minimal dose of 0.1 μM did not affect the viability of HG-treated cells, but loganin at 1 and 10 μM did increase the viability of HG-treated cells, incubated for 48 h. Although the data showed that both 1 and 10 μM of loganin could effectively improve 25-mM HG-induced cell death, we prefer to use the low concentration of loganin (1 μM) for the subsequent experiments. Of note, loganin at 50 μM decreased the cell viability of HG-treated cells (Figure 1B). To elucidate the direct effect of loganin on cell viability under NG conditions, we added various concentrations of loganin to NG-treated RSC96 cells, incubated for 48 h. Loganin significantly reduced cell viability at 50 μM, a level considered to induce direct cell toxicity (Figure 1C). Based on the above observations, 1-μM loganin incubation for 48 h was chosen for each subsequent experiment.

### 3.2. Loganin Diminished Intracellular ROS Generation in High-Glucose-Treated RSC96 Schwann Cells

To understand whether loganin affected the intracellular ROS levels induced by high glucose, 2′,7′-dichlorofluorescein-diacetate (DCFH_2_–DA) staining was performed. DCF fluorescence was measured after cells were incubated with 25-mM HG from 2 to 72 h using a fluorescence spectrophotometer. Intracellular ROS markedly increased at 4 h after 25-mM HG treatment, reached a plateau at 6 h and continued to accumulate from 24 to 72 h (Figure 2A). We also used flow cytometry to measure the intensity of DCF fluorescence and found increasing intensity after 25-mM HG treatment at 48 h (Figure 2B,C). To evaluate the antioxidant responses to 1-μM loganin in 25-mM-HG-treated RSC96 cells, the antioxidant N-acetylcysteine (NAC, 1 mM) was used as a positive control. Representative fluorescence images of RSC 96 cells are shown in Figure 2D. Figure 2E indicates the number of DCF positive cells [21]. DCF fluorescence significantly increased in the 25-mM HG group compared to the 5.6-mM NG group, and the response was strikingly reduced by 1-μM loganin, similar to 1-mM NAC (Figure 2F). Like NAC, the results suggested that loganin may protect against oxidative stress under 25-mM HG conditions. Figure 2G shows the quantitative overlapping results of DCF fluorescence intensity.

### 3.3. Loganin Attenuated Cell Death in High-Glucose-Treated RSC96 Schwann Cells

Previous studies found that 1-μM loganin reduced ROS generation; thus, we intended to clarify whether loganin can reduce cell death under high-glucose situations. Propidium iodide (PI) and Hoechst 33342 double staining assays were performed to estimate the ratio of dead cells. Hoechst 33342 dye penetrates living cells and PI stains dead cells only. Representative images of PI/Hoechst 33342 staining are shown in Figure 3A and quantification of cell death (percentage of dead cells of total cell number) is shown in Figure 3B. PI-positive cells increased in 25-mM-HG-treated RSC96 cells and decreased with 1-μM loganin or 1-mM NAC treatment. These results indicated that loganin or antioxidant NAC treatment reduced RSC96 cell death under high-glucose-mediated oxidative stress.

### 3.4. Loganin Decreased NLRP3 Inflammasome Assembly in High-Glucose-Treated RSC96 Schwann Cells

To further verify the possible mechanism of loganin’s effects on NLRP3 inflammasome assembly, RSC96 Schwann cells were pretreated with 1-μM loganin or 1-mM NAC for 2 h, then, co-incubated with 25-mM HG. RSC96 cells were divided into six groups with different treatments: normal glucose (NG, 5.6 mM), high glucose (HG, 25 mM), 25-mM HG with 1-μM loganin, 1-μM loganin, 25-mM HG with 1-mM NAC and 1-mM NAC. The protein expression levels of NLRP3 inflammasome (NLRP3, ASC and caspase-1) were determined. Compared with the 5.6-mM NG group, the expression of NLRP3, ASC and caspase-1 protein was significantly increased in the 25-mM HG group, but these proteins were downregulated by 1-μM loganin or 1-mM NAC treatment (Figure 4A,B). To further confirm the NLRP3 inflammasome assembly, we conducted cellular immunofluorescence assays. NLRP3 inflammasome was predominantly located in the cytoplasm. The merged images in Figure 4C, NLRP3 and ASC were colocalized (yellow), and the intensity was enhanced in the 25-mM HG group but attenuated in the 25-mM HG with 1-μM loganin group. Further, 25-mM HG stimulation caused NLRP3 and ASC to concentrate around the nucleus, and this effect may be blocked by 1-μM loganin. Likewise, the antioxidant 1-mM NAC inhibited the approach of ASC toward NLRP3 (Figure 4C). These results suggested that 1-μM loganin suppressed NLRP3 inflammasome assembly by ROS inhibition or scavenging activity.

### 3.5. Loganin Inhibited NLRP3 Inflammasome Activation in High-Glucose-Treated RSC96 Schwann Cells

NLRP3 is a key sensor of cellular stress. Canonical NLRP3 inflammasome activation occurs in two parallel and independent steps: the priming step initiated by NF-kB and the activation step triggered by a variety of molecular and cellular stimuli caused by ROS. To elucidate whether loganin inhibits NLRP3 inflammasome activation by blocking these two steps, Western blotting was performed to evaluate the influence of 1-μM loganin on NF-κB phosphorylation. 25-mM HG increased the ratio of phosphorylation of NF-κB and phosphorylation was remarkably decreased after 1μM loganin treatment (Figure 5A,B). To further determine whether loganin affected the activation and nuclear translocation of NF-κB, we used an immunofluorescent labeling method. As shown in Figure 5C, phospho-NF-κB fluorescence was co-localized with DAPI (nuclei marker) and accumulated in the nuclei of the 25-mM HG group but was decreased in 1-μM loganin treatment group. NF-κB activation contributes to the upregulation of *Nlrp3*, *Il-1β* and *Il-18* mRNA transcription. As shown in Figure 5D, 25-mM HG significantly increased the mRNA expression of *Nlrp3*, *Il-1β* and *Il-18* and this effect was significantly decreased by 1-μM loganin treatment. Additionally, ROS generation caused by high glucose affects the reaction of thioredoxin interacting protein (TXNIP) and purinergic receptor P2 × 7 (P2RX7) and then activates NLRP3 inflammasomes. As shown in Figure 5E,F, 25-mM HG increased TXNIP and P2RX7 expression and this expression was decreased by 1-μM loganin pretreatment. These results suggested that loganin may suppress both the priming and activating steps of NLRP3 inflammasomes in high-glucose-treated RSC 96 Schwann cells.

### 3.6. Loganin Reduced GSDMD-Mediated Pyroptosis in High-Glucose-Treated RSC96 Schwann Cells

To investigate the effect of loganin on GSDMD-mediated pyroptosis, the mRNA and protein expression of GSDMD, IL-1β and IL-18 were assessed by qRT-PCR (Figure 6A,B) and western blot (Figure 6C,D) and the levels of IL-1β secretion were assessed by ELISA assay (Figure 6B). The data showed that mRNA and protein levels of GSDMD, IL-1β and IL-18 were significantly enhanced in 25-mM-HG-treated RSC96 cells and decreased after 1-μM loganin treatment. Similarly, the levels of IL-1β secretion in cell supernatants after 25-mM HG treatment were remarkably higher than in the 5.6-mM NG, 25-mM HG + 1-μM loganin and 1-μM loganin alone groups. The expression of the N-terminal proteolytic fragment of GSDMD (GSDMD–NT) can translocate to the plasma membrane, forming a membrane pore, thereby triggering cell death. Western blotting data showed that GSDMD–NT was significantly increased in 25-mM-HG-treated RSC96 cells and significantly inhibited by 1-μM loganin treatment (Figure 6C,D). Additionally, GSDMD positive cells were tagged with the most powerful fluorescence on the plasma membrane of the HG group compared to the other groups (Figure 6E). These results suggested that loganin may suppress GSDMD-mediated pyroptosis in HG-treated RSC96 Schwann cells.

## 4. Discussion

High glucose levels have adverse effects on apoptosis, metabolism, proliferation and migration of Schwann cells [10]. The overproduction of ROS caused by high glucose induces oxidative stress and inflammation, a recognized mechanism in the pathogenesis of DPN [6,38]. This study provides the first evidence that loganin treatment reduces the activation of NLRP3 inflammasomes and subsequent pyroptosis by inhibiting ROS generation in high-glucose-treated RSC96 Schwann cells.

Loganin has been described as having a variety of pharmacological effects in many in vivo and in vitro studies [39]. Tseng et al. suggested that in primary mesencephalic neurons, loganin attenuates nerve cell apoptotic death, neurite damage and oxidative stress [32]. Babri et al. observed that acute administration of loganin could improve spatial memory in diabetic rats [40]. Wang et al. verified that loganin alleviates intestinal epithelial inflammation by regulating the TLR4/NF-κB and JAK/STAT3 signaling pathways [41]. Chao et al. exhibited that loganin has beneficial effects on Schwann cells by blocking Smad2 signaling induced by TNF-α [42]. Chen et al. found that loganin can reduce ROS and suppress the RAGE/p38 MAPK/NF-κB pathway to alleviate germ cell apoptosis in diabetes mellitus [24]. Loganin is a cornel iridoid glycoside (CIG) extracted from the fruits of *Cornus officinalis* [43]. Wang et al. demonstrated that CIG protects against white matter lesions induced by cerebral ischemia via the activation of BDNF and neuregulin-1 (NRG1)/ErbB4 pathways in the white matter [44]. Yuan et al. also confirmed that CIG has anti-inflammatory properties due to blockade of the STAT3/NF-κB pathway in the murine acute colitis model [45]. Li et al. revealed that loganetin from loganin enzymatically has been reported to protect against acute kidney injury by inhibiting the toll-like receptor 4 (TLR4) signaling pathway [46]. Our results show that RSC96 cell incubation with high glucose leads to cell death and intracellular ROS generation. In comparison with the antioxidant NAC, loganin treatment can reduce intracellular ROS generation and inhibit NF–κB–P2RX7–TNXIP protein expression from protecting against RSC96 cell injury caused by NLRP3 inflammasome activation. These findings are consistent with previous reports [32,45] that loganin protects against cells from oxidative stress, inflammation and apoptosis by regulating NF-κB related pathways. We further confirm that loganin could inhibit NLRP3 inflammasome activation from reducing high-glucose-induced RSC96 cell injury. Of note, the strength of loganin varies depending on the extraction method and the brand purchased, and the drug tolerance of different cells is also different. In our current study, 1 or 10-μM loganin treatment increased the viability of HG-treated RSC96 cells, but 50-μM loganin alone reduced cell viability in normal glucose-treated RSC 96 cells and worsened cell viability after high-glucose treatment.

NLRP3 plays an important role in the development of diabetes. NLRP3 inflammasomes contain the sensor molecule NLRP3, an apoptosis-associated speck-like protein containing a caspase recruitment domain CARD (ASC) and pro-caspase-1. It is widely known that NLRP3 inflammasomes are associated with the pathogenic mechanisms of type 2 diabetes and its associated complications [22,47,48]. Ding et al. found that the regulation of NLRP3 can affect endoplasmic reticulum stress to regulate glucose tolerance, insulin resistance, inflammation and apoptosis in adipose tissue in type 1 and type 2 diabetes [47]. Gan et al. revealed that high glucose induced the loss of retinal pericytes partly via NLRP3-caspase-1-GSDMD-mediated pyroptosis [20]. Yu et al. also found that NLRP3 inflammasome activation and pyroptosis was involved in the development and progression of diabetes [22]. Jia et al. found that paclitaxel-elicited mitochondria damage and reactive oxygen species production may result in activation of NLRP3 inflammasomes in peripheral nerve [49]. Our results exhibited that loganin could reduce NLRP3 inflammasome activation, procaspase-1 cleavage and ASC upregulation in RSC96 cells after high-glucose stimulation. Thus, we suggest that loganin may have the ability to improve HG-induced metabolic disorders.

Two signals are required to complete NLRP3 inflammasome activation. First, a priming signal induced by NF-κB is required for the upregulation of NLRP3, pro-IL-1β and pro-IL-18. Second, activation signals activated by pathogen-associated molecular patterns (PAMPs) and damage-associated molecular patterns (DAMPs) trigger assembly into the NLRP3 inflammasome complex, after which activation of caspase-1 is responsible for pyroptotic cell death [50,51,52]. An et al. demonstrated that the activation of ROS/MAPKs/NF-κB/NLRP3 in osteoclasts can cause osteoporosis in diabetes [53]. Zhong et al. found that NF-κB primes the NLRP3-inflammasome for activation by inducing pro-IL-1β and NLRP3 expression [54]. NF-κB not only initiates inflammation by increasing the production of inflammatory cytokines, chemokines and adhesion molecules, but also regulates cell proliferation, apoptosis, morphogenesis, differentiation and inflammasome activation [55]. Thioredoxin interacting protein (TXNIP) belongs to the thioredoxin (TRX) system and is an important regulator of oxidative stress. TRX is the main molecule that resists oxidative stress in cells. Thus, increased oxidative stress will reduce TRX activity and increase TXNIP performance [56]. Zhou et al. found that NLRP3 interacted with TXNIP. ROS induced the dissociation of TXNIP from TRX and allowed it to bind to NLRP3. TXNIP activates the NLRP3 inflammasome and subsequent secretion of IL-1β in the pathogenesis of type 2 diabetes [57]. P2RX7 is a ligand-gated ion channel belonging to the purinergic type 2 receptor family (P2). P2RX7 is highly expressed on immune cells such as macrophages, mast cells and microglial cells and can also be found on oligodendrocytes. P2RX7 has high relevance under pathological conditions, activated by high millimolar ATP concentrations [58]. P2RX7 stimulation represents the second signal to inflammasome activation by triggering K^+^ efflux, as ATP-induced P2RX7 activation causes a sustained increase in intracellular Ca^2+^, inflammasome assembly and subsequent caspase-1 activation [59,60]. Excessive activation of P2RX7 and NLRP3 leads to increased inflammatory cytokine secretion, such as IL-1β in depression and diabetes [61]. Colomar et al. pointed out that the maturation and release of interleukin 1β is required to stimulate P2RX7 in lipopolysaccharide-primed mouse Schwann cells [38]. Gu et al. revealed that TXNIP regulates the mechanism of NLRP3 inflammasome activation in diabetic nephropathy [39,40]. Our results showed the priming signal as phospho-NF-κB translocated to the cell nucleus and upregulated NLRP3, IL-1β and IL-18 gene expression. Increases of P2RX7 and TXNIP activate the NLRP3 inflammasome assembly. In this study, we observed that loganin could inhibit the formation and activation of NLRP3 inflammasome due to its retardation of oxidative stress and the expression of NF–κB–P2RX7–TNXIP proteins (Figure 7). Pyroptosis is a pathway of cell death that inherently results in inflammation. The activation state or differentiation state of individual cells may determine the dominant death pathway [62,63]. Pyroptosis is a gasdermins-mediated pro-inflammatory programmed cell death, which has been widely studied in inflammatory disease models [64]. Both caspase-1-dependent and independent pyroptosis lead to the release of IL-1β and IL-18. IL-1β and IL-18 are effective proinflammatory cytokines, exaggerating inflammation by inducing expression of other proinflammatory cytokines and adhesion molecules [65]. Previous reports found that GSDMD is the executor of pyroptosis. GSDMD is cleaved by caspase-1 to the N-terminal proteolytic fragment of GSDMD (GSDMD–NT), forming pores in the cell membrane to progressively trigger cell death, releasing pro-inflammatory cytokines such as IL-1β and IL-18 to activate a strong inflammatory response [10,66,67,68]. We also found that activation of caspase-1 cleaves GSDMD to generate GSDMD–NT and stimulation of IL-1β release, resulting in pyroptosis. Based on our findings, loganin may improve pyroptosis via oxidative stress mitigation in RSC96 cells under high-glucose exposure.

In conclusion, loganin attenuation of pyroptosis could be due to the reduction of intracellular ROS and inhibition of NF-κB and NLRP3 inflammasome activation (Figure 7). From our in vitro findings, we suggest that loganin may have therapeutic potential in hyperglycemia–induced diabetic peripheral neuropathy.

## Figures and Tables

**Figure 1 cells-09-01948-f001:**
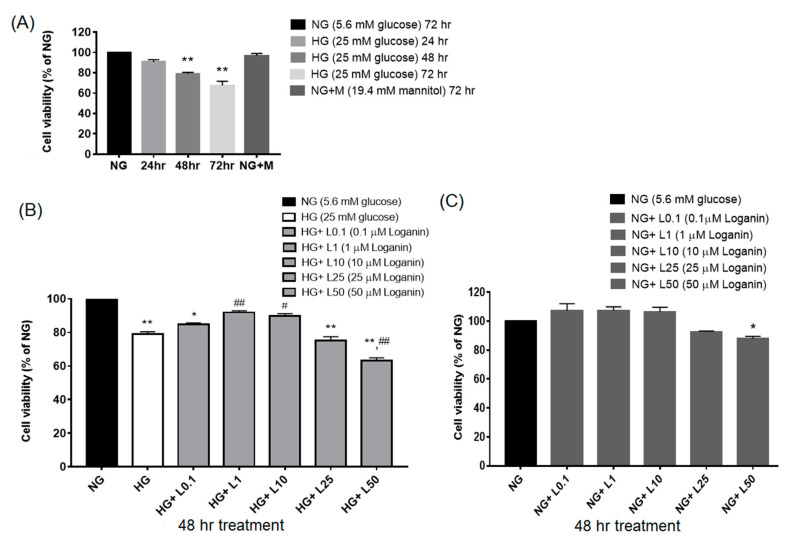
Effect of high glucose (HG) and loganin on the cell viability of rat RSC96 Schwann cells by Cell Counting Kit-8 (CCK-8) assay. (**A**) RSC96 cells were exposed to 25-mM HG for 24, 48 and 72 h. 5.6-mM NG + 19.4-mM mannitol for 72 h incubation was used as an osmotic control. * *p* < 0.05, ** *p* < 0.01, compared with 5.6-mM normal glucose (NG); (**B**) The effect of different concentrations (0.1, 1, 10, 25, 50 μM) of loganin was incubated for 48 h on the viability of 25-mM-HG-treated RSC96 cells; (**C**) effect of different concentrations of loganin was incubated for 48 h on the viability of 5.6-mM-NG-treated RSC96 cells. * *p* < 0.05 and ** *p* < 0.01 vs. normal glucose (NG); ^#^
*p* < 0.05 and ^##^
*p* < 0.01 vs. high glucose (HG).

**Figure 2 cells-09-01948-f002:**
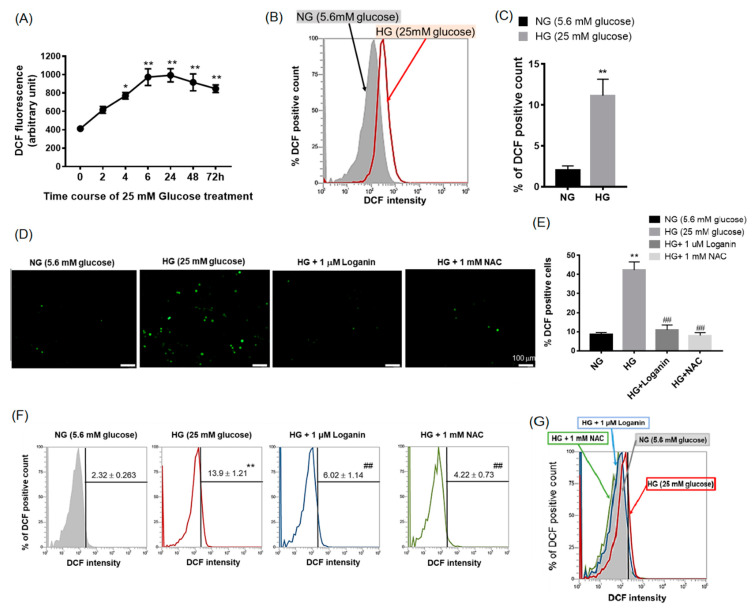
Effect of high glucose and loganin on intracellular ROS generation in rat RSC96 Schwann cells assayed by 2′,7′-dichlorofluorescein–diacetate (DCFH_2_–DA) staining. (**A**) Intracellular reactive oxygen species (ROS) levels of 25-mM-HG-treated RSC96 cells in 96-well plates were measured by the fluorescence intensity using a fluorescence spectrophotometer; (**B**) fluorescence intensity measured by flow cytometry. Overlay flow cytometry histograms represent DCF fluorescence in the 25-mM-HG-treated group (red line) and 5.6-mM NG (gray area); (**C**) quantitative analysis of DCF fluorescence intensity. To evaluate the antioxidant effect of loganin, N-acetyl-L-cysteine (NAC) antioxidant was added as a control group. RSC96 cells were incubated with 1-μM loganin or 1-mM N-acetylcysteine for 48 h; (**D**) Images were obtained using a fluorescence microscope (magnification, ×100; scale bar, 100 µm) with DCF-positive cells in green and (**E**) reflected as positive cells in each view (mean ± SEM, *n* = 6); (**F**) fluorescence intensity measured by flow cytometry; (**G**) overlay flow cytometry histograms of cultures. An increase in DCF fluorescence was observed in the HG-treated group (red line) with no change in 25-mM HG with 1-μM loganin (blue line) or 1-mM NAC (green line) treatment, compared with 5.6-mM NG (gray area). Values represent triple-repeated experiments and average DCF fluorescence intensity fold change relative to the NG group. * *p* < 0.05 and ** *p* < 0.01 vs. NG group; ## *p* < 0.01 vs. HG group.

**Figure 3 cells-09-01948-f003:**
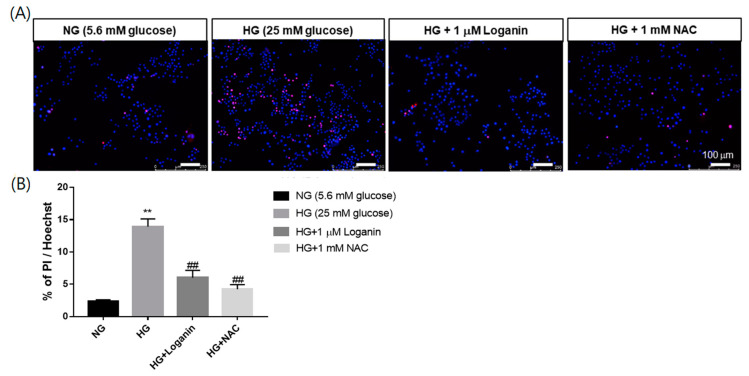
Effect of loganin on cell death in high-glucose-treated rat RSC96 Schwann cells by PI/Hoechst 33342 assay. RSC96 cells were pretreated with 1-μM loganin or 1-mM N-acetylcysteine and incubated with 25-mM-high glucose (HG) for 48 h. (**A**) Representative composite images were acquired by fluorescence microscope (magnification, ×100; scale bar, 100 µm). Dead cells were stained using PI and shown in red, and nuclei were stained with Hoechst 33342 (blue); (**B**) percentages of cell death were calculated by determining the ratio of PI-positive cells to Hoechst 33342-stained cells. Bars indicate mean ± SEM of the mean (*n* = 6). ** *p* < 0.01 vs. NG group; ^##^
*p* < 0.01 vs. HG group.

**Figure 4 cells-09-01948-f004:**
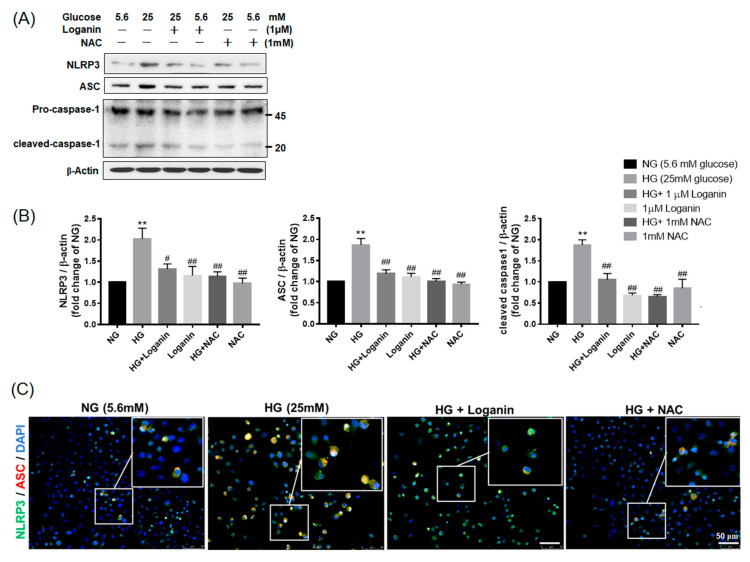
Loganin effects on NOD-like receptor protein 3 (NLRP3) inflammasome formation in high-glucose-treated rat RSC96 Schwann cells. (**A**) Western blot images show the expression of NLRP3, ASC and pro-/cleaved-caspase-1 protein in high-glucose-treated cells at 48 h; (**B**) relative fold changes of NG corresponding to (A). ** *p* < 0.01 vs. normal glucose (NG, 5.6 mM); ^#^
*p* < 0.05 and ^##^
*p* < 0.01 vs. HG group. Data expressed as mean ± SEM, *n* = 3; (**C**) double-immunofluorescent labeling with NLRP3 (green) and ASC (red). Nuclei counterstained with DAPI. Merge revealed colocalization (yellow) in cells (inset). Magnification, ×200; scale bar, 50 µm.

**Figure 5 cells-09-01948-f005:**
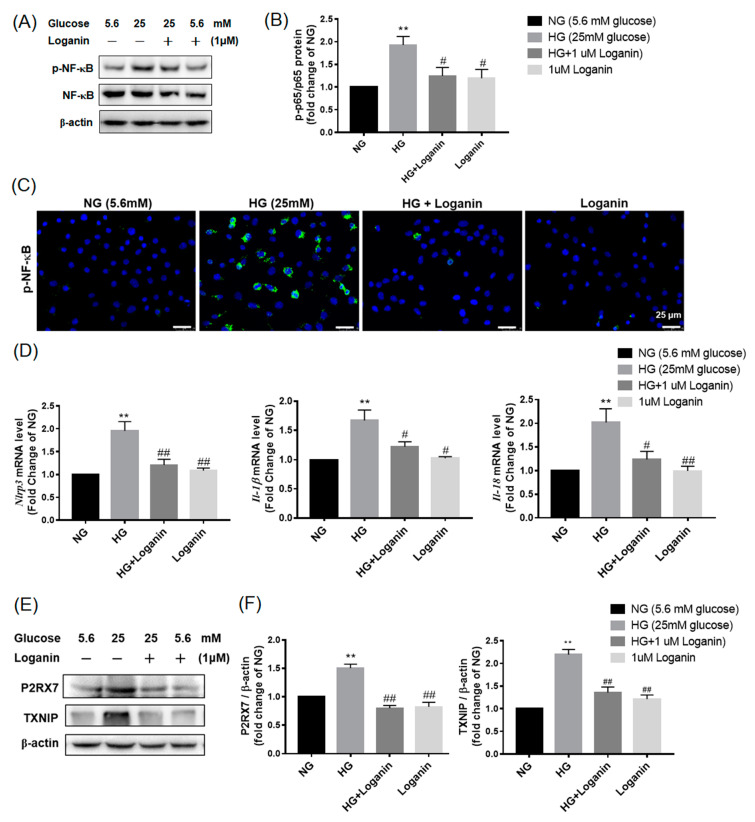
Loganin inhibited the priming and activating steps of NLRP3 inflammasome formation in high-glucose-treated rat RSC96 Schwann cells. (**A**) Western blot images show the expression of phospho-NF-κB and NF-κB protein in 25-mM-HG-treated cells with 1-μM loganin and 1-mM NAC at 48 h; (**B**) relative fold changes of 5.6-mM NG corresponding to (**A**); (**C**) immunofluorescent labeling with phospho-NF-κB translocation. Nuclei counterstained with DAPI. Merge revealed colocalization in cells. Magnification, ×400; scale bar, 25 µm; (**D**) relative mRNA levels of NLRP3, IL-1β and IL-18 were quantified; (**E**) western blotting analysis of P2RX7 and TXNIP in each treatment; (**F**) Data expressed as mean ± SEM, *n* = 3. ** *p* < 0.01 vs. normal glucose (NG, 5.6 mM); ^#^
*p* < 0.05 and ^##^
*p* < 0.01 vs. HG group.

**Figure 6 cells-09-01948-f006:**
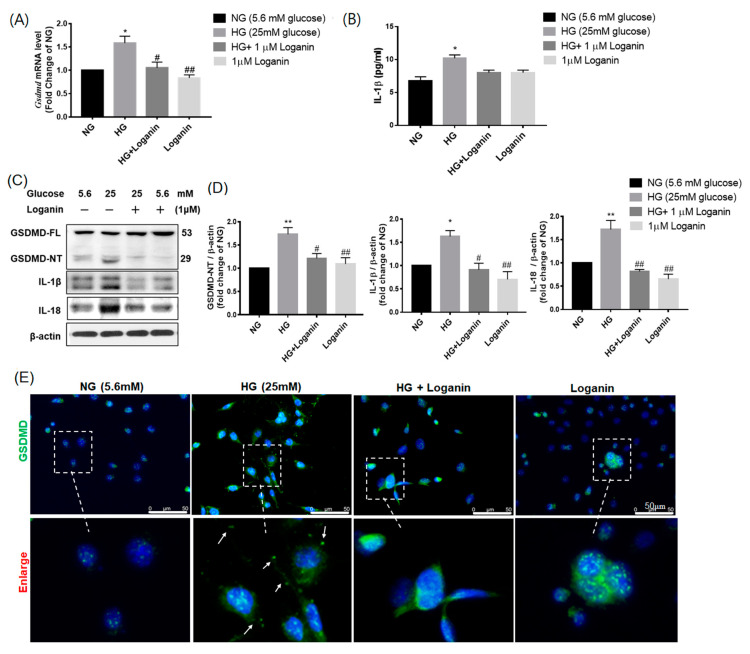
Loganin effects on gasdermin D (GSDMD)-mediated pyroptosis in high-glucose-treated rat RSC96 Schwann cells. (**A**) Quantification of the relative value of GSDMD mRNA levels by real-time PCR; (**B**) The amount of IL-1β released into the culture medium was measured by ELISA; (**C**) western blot images show the expression of GSDMD-full length, GSDMD-N terminal (NT), IL-1β and IL-18 protein in each treatment of RSC96 cells at 48 h and (**D**) represents the fold change corresponding to (**C**). * *p* < 0.05 and ** *p* < 0.01 relative to NG; ^#^
*p* < 0.05 and ^##^
*p* <0.01 relative to the HG group. The data are expressed as ± SEM, *n* = 3 (**E**) immunofluorescent labeling with GSDMD. Nuclei counterstained with DAPI. In the locally enlarged lower panel, the arrow shows that GSDMD relocated to the cell membrane. Magnification, ×400; scale bar, 50 µm.

**Figure 7 cells-09-01948-f007:**
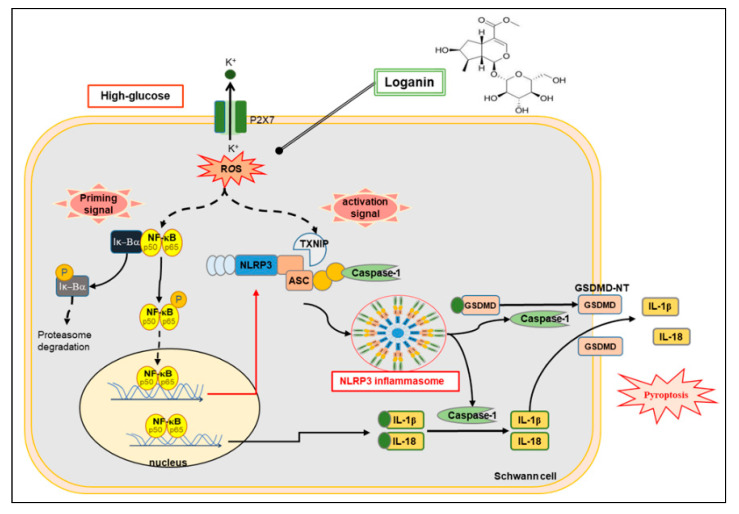
Schematic diagram summarizing the proposed mechanism by which loganin attenuates pyroptotic death in high-glucose-induced rat RSC96 Schwann cells by inhibiting ROS generation and NLRP3 inflammasome activation. ROS—reactive oxygen species; NLRP3—NOD-like receptor protein 3; NF-κB—nuclear factor κB; P2RX7—purinergic receptor P2 × 7; TXNIP—thioredoxin-interacting protein; ASC—apoptosis-associated speck-like protein containing a caspase recruitment domain; GSDMD—gasdermin D; GSDMD–NT—gasdermin D-N-terminal; IL-1β—interleukin-1beta; IL-18—interleukin-18.

**Table 1 cells-09-01948-t001:** The primer sequences used for qPCR assay.

*Gene*	Accession No.	Forward Primer (5′-3′)	Reverse Primer (5′-3′)
*Nlrp3*	NM_001191642.1	CGGTGACCTTGTGTGTGCTT	TCATGTCCTGAGCCATGGAAG
*Gsdmd*	NM_001130553.1	AAGATCGTGGATCATGCCGT	CTCAGGAGGCAGTAGGGCTT
*Il-1 β*	NM_031512.2	AAATGCCTCGTGCTGTCTGA	AGGCCACAGGGATTTTGTCG
*Il-18*	NM_019165.1	ACCACTTTGGCAGACTTCACT	ACACAGGCGGGTTTCTTTTG
*β-actin*	NM_031144.3	GACCCAGATCATGTTTGAGACC	AGGCATACAGGGACAACACA

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
