# Peer review of "Loganin Attenuates High Glucose-Induced Schwann Cells Pyroptosis by Inhibiting ROS Generation and NLRP3 Inflammasome Activation"

_cells, 2020, doi:10.3390/cells9091948_

Round 1
Reviewer 1 Report
This study investigated how the Loganin attenuates high glucose-induced Schwann cells pyroptosis by inhibiting ROS generation and NLRP3 inflammasome activation.
The rational behind the experiment was clear and straight forward. The manuscript is almost well written
While many different sources are used to set up the study in the introduction, little previous evidence is stated. The introduction is thus short and poorly sets up the rationale for the study.
More attention to how this study fits into previous work during the relationship between inflammation and NLRP3 activation should be added to improve this section. For example 10.18632/oncotarget.25823 or 10.3390/ijms21062144
There are some minor grammar issues that should be fixed in order to aid the accessibility of the results to the reader.
Please provide better quality of all images presented in particularly figures 2, 4, 5 and 6.
The discussion does a good job at explaining the importance of the results in the context of the inflammatory pathways involved. However, incorporation of previous results from other related studies is lacking such as 10.1371/journal.pone.0176965 Or 10.5681/apb.2013.015 or 10.1111/bph.14595 or 10.2147/NDT.S228417.
Incorporating comparisons with other studies would increase the strength of the paper. Please describe better the role of inflammation during diabetic peripheral neuropathy in in vivo models (please refer for example 10.1096/fj.201900538R)
Author Response
The authors would like to take this chance to thank you for your positive supports and comments on our work. We would like to respond to your concerns by point-to-point as follows.
- The rationale behind the experiment was clear and straightforward. The manuscript is almost well written.
Reply: Thank you very much for your overall positive view of our work.
- While many different sources are used to set up the study in the introduction, little previous evidence is stated. The introduction is thus short and poorly sets up the rationale for the study.
Reply: We appreciated your comments. In the introduction section, we have added some additional information and relevant evidence to support the rationale or design of our study. Please see lines 49-53 for details.
- More attention to how this study fits into previous work during the relationship between inflammation and NLRP3 activation should be added to improve this section. For example, 10. 18632/oncotarget.25823 or 10.
Reply: Thank you for your suggestion. We have included these 2 relevant papers in the introduction section. Please see lines 68-70 for details.
- There are some minor grammar issues that should be fixed in order to aid the accessibility of the results to the reader.
Reply: Thank you so much. We have fixed the grammar problem in this revised manuscript and this article has been edited by Mr. Gary Mawyer, a professional editor recently retired from the University of Virginia.
- Please provide better quality of all images presented in particularly figures 2, 4, 5 and 6.
Reply: As per your suggestion, we have provided the high quality of images as you mentioned.
- The discussion does a good job at explaining the importance of the results in the context of the inflammatory pathways involved. However, incorporation of previous results from other related studies is lacking such as 10.1371/journal.pone.0176965 or 10.5681/apb.2013.015 or 10.1111/bph.14595 or 10.2147/NDT.S228417.
Reply: We agreed with your insightful suggestion. We have included some relevant articles as you suggested and added the descriptions in the discussion (Line 352-364, highlighted text in red). Please take a quick look, thanks.
- Incorporating comparisons with other studies would increase the strength of the paper. Please describe better the role of inflammation during diabetic peripheral neuropathy in in vivo models (please refer for example 1096/fj.201900538R)
Reply: Thank you for your invaluable suggestion. We have included the article and added some relevant information to support the purpose of our study. Please see lines 49-53 for details.
Reviewer 2 Report
This study evaluated the neuroprotective effect of loganin on high glucose induced rat Schwann cell line RSC96 injury.The rational behind the experiment was clear and straight forward. The manuscript is almost well written.
While many different sources are used to set up the study in the introduction, little previous evidence is stated. The introduction is thus short and poorly sets up the rationale for the study. More attention to how this study fits into previous work in NLRP3 inflammasome and inflammation should be added to improve this section. Please refer to doi: 10.3390/ijms21124223
Please improve the image quality.
There are some minor grammar issues that should be fixed in order to aid the accessibility of the results to the reader.
The discussion does a good job at explaining the importance of the results in the context of the inflammatory pathways involved. However, incorporation of previous results from other related studies is lacking. Incorporating comparisons with other studies would increase the strength of the paper. Please describe better the role of the NLRP3 inflammasome in the inflammation pathway. Please refer to: doi: 10.3390/ijms21062144, 10.1096/fj.201601349R.
Author Response
The authors would like to thank the reviewer’s positive support for our work. Our point-to-point responses are listed below.
- While many different sources are used to set up the study in the introduction, little previous evidence is stated. The introduction is thus short and poorly sets up the rationale for the study. More attention to how this study fits into previous work in NLRP3 inflammasome and inflammation should be added to improve this section. Please refer to doi: 10.3390/ijms21124223
Reply: Thank you for your invaluable comments. We have included this review article in the introduction section. Besides, we have added some additional information and relevant evidence to support the rationale or design of our study. Please see lines 49-53 for details.
- Please improve the image quality.
Reply: Sure, we have provided the high quality of images as you suggested.
- There are some minor grammar issues that should be fixed in order to aid the accessibility of the results to the reader.
Reply: Thank you for your suggestion. We have fixed the grammar problem in this revised manuscript and this article has been edited by Mr. Gary Mawyer, a professional editor recently retired from the University of Virginia.
- The discussion does a good job at explaining the importance of the results in the context of the inflammatory pathways involved. However, incorporation of previous results from other related studies is lacking such as 10.1371/journal.pone.0176965 or 10.5681/apb.2013.015 or 10.1111/bph.14595 or 10.2147/NDT.S228417.
Reply: We agreed with your insightful suggestion. We have included some relevant articles as you suggested and added the descriptions in the discussion (Line 352-364, highlighted text in red). Please take a quick look, thanks.
Reviewer 3 Report
The manuscript has an interesting idea but the design and results are not properly described and the results is that the manuscript it is very confusing.
There are some English editing to be done (line 54 - factore? lines 186-187 "To clarify the effect of high glucose (HG) on RSC96 cell viability, and 25 mM glucose was used to mimic the diabetic situation. "???)
2.2. Cell viability
Why was the cell viability evaluated in a completely different model compared to the one used at 2.1. Evaluating the cell viability for cells exposed to loganin + 25mM glucose is not relevant for an experiment evaluating the effect of loganin pretreatment on cells exposed to normal glucose/high glucose/glucose+ mannitol.
Also, the evaluation of ROS was performed in cells exposed to 25mM glucose with/without loganin. Yet another type of model.
Line 114 - 10L???
2.4. ELISA
What was the cell model used for evaluating IL1beta? Pretreatment with loganin or co-treatment glucose+loganin?
Same question for Western blot and all the other experiments in the manuscript.
2.6. Cell death
What type of treatment? Why use different types of plates for experiments? Also different cell densities?
3.1.
In the Cell viability assay section authors describe exposing cells to high glucose, yet in the results section exposure to manitol and high glucose is mentioned. In figure 1 results are presented also for cell exposure to loganin alone. It is confusing.
In section 2.1. authors mention exposing ells to 0,1, 1 and 10 microM loganin, in 3.1. results are presented not only for the above mentioned concentrations but also for 25 and 50microM.
Figure 1 How long were the cells exposed to NG+M???
3.2.
Why choose 1microM loganin for ROS assay, since the previous experiment showed significant results for both 1 and 10microM. Same question for following experiments.
NG+M control was used in cell viability; why not in ROS experiment? Same question for following experiments.
Discussion
Pretreatment? Co-treatment ? The whole discussion section should be re-written after clarifying the experimental design and after creating an unified model for all the manuscript regarding the way the experiments were conducted.
Author Response
The authors appreciate your time and effort to go through our paper and to give us some helpful suggestions. As follows, our responses are the point to point as you concerned.
- There are some English editing to be done (line 54 - factore? lines 186-187 "To clarify the effect of high glucose (HG) on RSC96 cell viability, and 25 mM glucose was used to mimic the diabetic situation. "???)
Reply: All typos and sentences have been corrected, thanks.
- 2. Cell viability
Why was the cell viability evaluated in a completely different model compared to the one used at 2.1. Evaluating the cell viability for cells exposed to loganin + 25mM glucose is not relevant for an experiment evaluating the effect of loganin pretreatment on cells exposed to normal glucose/high glucose/glucose+ mannitol.
Reply: Thank you for your comment. The group of normal glucose (NG)+loganin wanted to see the direct effect of loganin on the viability of NG-treated cells. It is to realize the direct effect of loganin on cell viability under NG conditions. Thus, it is easy to clarify how loganin modulates high glucose (HG)-induced cell injury or death. In this revised manuscript, we separated loganin’s data on the viability of RSC96 cells under NG and HG conditions.
Line 114 - 10L???
Reply: Typos have been fixed.
- 4. ELISA
What was the cell model used for evaluating IL1beta? Pretreatment with loganin or co-treatment glucose+loganin?
Same question for Western blot and all the other experiments in the manuscript.
Reply: Loganin was pretreated and then incubated in normal or high glucose conditions. All experiments were performed and described in the 2.1. cell culture and drug treatment.
- 6. Cell death
What type of treatment? Why use different types of plates for experiments? Also different cell densities?
Reply: 1) All experiments were pretreated with loganin or NAC.
2) We use different culture dishes and plates to meet the needs of different experiments
and to fit the requirements of each equipment.
3) Cell densities: 2*105 cells in 10 cm dish; 1*104 cells/well in 24 well plate
2*103 cells/well in 96 well plate.
- 1. In the cell viability assay section authors describe exposing cells to high glucose, yet in the results section exposure to manitol and high glucose is mentioned. In figure 1 results are presented also for cell exposure to loganin alone. It is confusing.
Reply: Thank you for your comment. To exclude the osmotic effects caused by high concentrations of glucose, thus, mannitol is commonly used as an osmotic control. In this study, there were no significant differences between NG with mannitol and NG groups found, and therefore the osmotic effects could be excluded.
In section 2.1. authors mention exposing ells to 0,1, 1 and 10 microM loganin, in 3.1. results are presented not only for above mentioned concentrations but also for 25 and 50 microM.
Reply: Thanks, we have fixed this matter.
Figure 1, How long were the cells exposed to NG+
Reply: 72 hr, we have indicated in the bar chart.
- Why choose 1 microM loganin for ROS assay, since the previous experiment showed significant results for both 1 and 10 microM. Same question for following experiments.
NG+M control was used in cell viability; why not in ROS experiment? Same question for following experiments.
Reply: In cell viability experiments, the influence of time course, drug concentrations under high glucose situations was monitored. Figure 1 showed that both 1 and 10 microM of loganin can effectively improve high glucose-induced cell injury or death, thus, we prefer to pick and choose the low concentration of loganin (1 microM) for the subsequent experiments. Moreover, we also observed that the osmotic agent (NG+mannitol) little affect the cell viability, and accordingly the osmotic control did not monitor in the following experiments. We hope our responses can satisfy your concerns.
- Discussion
Pretreatment? Co-treatment? The whole discussion section should be re-written after clarifying the experimental design and after creating an unified model for all the manuscript regarding the way the experiments were conducted.
Reply: We have rechecked and fixed, thanks a lot.
Reviewer 4 Report
- Line 48: are the authors referring to global prevalence here? Please clarify.
- In reference to the statement that many studies have confirmed the role of loganin in DPN treatment; authors cite only one reference. This should be edited.
- Authors mention (line 191) that 25mM glucose was added to mimic diabetic conditions. They should discuss the correlation between 25mM glucose and physiological hyperglycemia in humans and cite appropriate reference(s).
- How long were the cells treated with NG+mannitol (Fig 1A)? How long were the cells treated with loganin (Fig 1B and C)? These should be specified in narrative as well as in figure legend.
- X-axis labels in Fig 2E are not clear. What are HL and L?
- Lack of consistency in immunocytochemistry information. Line 275 states, ‘NLRP3 and ASC (yellow)” while Line 286 says ‘NLRP3 and ASC 286 (red)’. It will be better if the authors state in the methods that alexa fluor 488 and 594 are green and red fluorescent dyes, respectively.
Incomplete sentences/grammatical errors/syntactic errors:
Line 49: The development and progression of diabetic neuropathy attributed to hyperglycemia and metabolic disorder.
Line 57: Should rephrase as: “there are two types of Schwann cells….”
Lines 67-69: Doesn’t make sense the way it is written: “Some chronic inflammatory diseases, such as endometriosis, bronchiolitis obliterans syndrome, etc., have also been found to reduce NLRP3 activation if the upstream Formyl Peptide Receptor 1 (Fpr-1) is inhibited”
Line 109: “concentration” should be changed to “concentrations”
Line 203-205: To realize the direct effect of loganin on cell viability under NG conditions, the NG+loganin group designed to observe the impact of loganin on NG-treated RSC96 cells.
Fig 6D is wrongly labeled as 6E.
Author Response
Response to Reviewer #4
Thank you for your comments and suggestions on our work, which have enabled us to improve this manuscript. We are happy to respond to your opinions and have made the necessary amendments to the final revision.
- Line 48: are the authors referring to global prevalence her? Please clarify.
Response: Sure, we have revised it, thanks.
- In reference to the statement that many studies have confirmed the role of loganin in DPN treatment; authors cite only one reference. This should be edited.
Response: We have cited 3 references. Please take a quick look on page 2, Line 86.
- Authors mention (line 191) that 25 mM glucose was added to mimic diabetic conditions. They should discuss the correlation between 25 mM glucose and physiological hyperglycemia in humans and cite appropriate reference(s).
Response: Thank you for your invaluable comments. As per your suggestion, we have added the physiological levels of normal and high blood glucose in this revised manuscript, and the references are also included. Our detailed descriptions are as follows. The American Diabetes Association defined an average fasting plasma glucose level < 5.6 mM; hyperglycemia reaches the glucose level >22.2–25 mM [34]. To simulate an uncontrolled diabetic state, we designed to culture the cells in 25 mM glucose and investigated the effect of high glucose on the viability of RSC96 cells. The 5.6 mM glucose medium is close to physiological levels [34-37].
- How long were the cells treated with NG+mannitol (Fig 1A)? How long were the cells treated with loganin (Fig 1B and 1C)? These should be specified in narrative as well as in figure legend.
Response: Thanks for your suggestions; we have added the incubation time in the narrative (Line 197, 202, 208) as well as in figure legend (Line 214, 216, 217).
- X-axis labels in Fig 2E are not clear. What are HL and L?
Response: Thanks, we have fixed the problems. HL and L have been changed as HG+loganin and HG+NAC, respectively.
- Lack of consistency in immunocytochemistry information. Line 275 states, “NLRP3 and ASC (yellow)” while Line 286 says “NLRP3 and ASC (red)”. It will be better if the authors state in the methods that Alexa fluor 488 and 594 are green and red fluorescent dyes, respectively.
Response: Thanks for your insightful suggestions. We have fixed the lack of consistency in immunocytochemistry information.
Line 275 (was): Now is Line 279, we have corrected the sentence as follows. The merged images in Fig. 4C, NLRP3 and ASC were co-localized (yellow), and…
Line 286 (was): Now is Line 290, we have fixed as NLRP3 (green) and ASC (red).
As per your suggestion, we have added the sentence (Alexa fluor 488 and 594 are green and red fluorescent dyes, respectively.) in the Methods (Line 172).
Incomplete sentences/grammatical errors/syntactic errors:
Line 49: The development and progression of diabetic neuropathy attributed to hyperglycemia and metabolic disorder
Response: We have corrected the sentence as follows. It is generally agreed that hyperglycemia mainly contributes to the development of diabetic neuropathy or DPN.
Line 57: Should rephrase as: “there are two types of Schwann cells…”
Response: We have fixed the sentences as follows. There are two types of Schwann cells: myelinated and unmyelinated. Both types of Schwann cells are specialized glial cells in the peripheral nerve system.
Lines 67-69: Doesn’t make sense the way it is written: “Some chronic inflammatory diseases, such as endometriosis, bronchiolitis obliterans syndrome, etc., have also been found to reduce NLRP3 activation if the upstream Formyl Peptide Receptor 1 (Fpr-1) is inhibited [16, 17].
Response: We have revised the sentences as follows. Some chronic inflammatory conditions, such as endometriosis, bronchiolitis obliterans syndrome, etc. have been shown to regulate NLRP3 inflammasome via the Formyl Peptide Receptor 1 (Fpr-1)-mediated signaling pathway [16, 17].
Line 109: “concentration” should be changed to “concentrations.”
Response: We have fixed it, thanks.
Line 203-205: To realize the direct effect of loganin on cell viability under NG conditions, the NG+loganin group designed to observe the impact of loganin on NG-treated RSC96 cells.
Response: We have rewritten the sentence as follows. To elucidate the direct effect of loganin on cell viability under NG conditions, we added various concentrations of loganin to NG-treated RSC96 cells, incubated for 48 hr. Please see Lines 204-206 for details, thanks.
Fig 6D is wrongly labeled as 6E.
Response: We have fixed the problem, thanks.
Round 2
Reviewer 3 Report
Introduction
“ The development and progression of diabetic neuropathy are attributed to hyperglycemia, metabolic disorder, which leads lead to neuroinflammation, peripheral nervous tissue changes, decreases decrease in the speed of nerve conduction, and altered activity of the nerve fiber enzyme Na+, K+-ATPase, which provokes provoke reactive oxygen species [1] release”
2.2. Cell viability assays
„Then cells were cultured in medium with 25 mM high glucose and 0.1, 1, and 10 μM loganin for 24, 48, and 72 hrs”
No justification was providee for the fact that the viability of cells was tested, not only in a different cell model but also with different stimulus concentration (experiments were performed for 0.1, 1, 10, 25, 50 μM loganin).
Such details should be included in each chapter of the manuscript.
Results
„To clarify the effect of high glucose (HG) on RSC96 cell viability, and??? 25 mM glucose was used to mimic the diabetic situation. „ – English???
„Loganin alone significantly reduced cell viability at >50 μM”??? The highest concentration that was tested was 50 μM so no conclusion could be drawn for exposure to higher concentrations
The discussion section of the manuscript was changed only using data from literature. Authors should improve this section with details regarding the design of their own experiment and justifications for the results, according to previous suggestions.
Authors should understand that previous comments are relevant and the responses should be included into the manuscript!!! Just providing the reviewer with some responses is not enough. Those responses are VERY relevant for the manuscript!
Author Response
The authors would like to take this chance to thank you for your comments on our work. Your comments have enabled us to improve this manuscript. We are pleased to follow the suggestions and have made the necessary amendments for final revision.
- Introduction
“The development and progression of diabetic neuropathy are attributed to hyperglycemia, metabolic disorder, which leads lead to neuroinflammation, peripheral nervous tissue changes, decreases decrease in the speed of nerve conduction, and altered activity of the nerve fiber enzyme Na+, K+-ATPase, which provokes provoke reactive oxygen species release [1]”
Response: Thank you very much for your help in revising this sentence. We have separated this long sentence into two sentences: The development and progression of diabetic neuropathy attributed to hyperglycemia and metabolic disorder. DPN caused neuroinflammation, peripheral nerve injury, reduced nerve conduction velocity, and altered the nerve fiber Na+, K+-ATPase activity, which provokes reactive oxygen species (ROS) release [3]. Also, you can see lines 49-52 for details.
- 2. Cell viability assays
“Then cells were cultured in medium with 25 mM high glucose and 0.1, 1, and 10 μM loganin for 24, 48, and 72 hrs”
No justification was provided for the fact that the viability of cells was tested, not only in a different cell model but also with different stimulus concentration (experiments were performed for 0.1, 1, 10, 25, 50 μM loganin).
Such details should be included in each chapter of the manuscript.
Response: Thank you for your invaluable comments. As per your suggestion, we have included some relevant descriptions throughout this revised manuscript where needed. The main conceptions we added, please see page 5, last paragraph with highlighted text. We considered that the revised manuscript had provided relevant information to meet your concerns. So far, this manuscript should be precise and comprehend for the readers.
- Results
“To clarify the effect of high glucose (HG) on RSC96 cell viability, and??? 25 mM glucose was used to mimic the diabetic situation „ – English???
Response: We have revised this sentence as follows: To mimic diabetic situations, we used 25 mM glucose to investigate the effect of high glucose (HG) on the viability of RSC96 cells (Lines 190-191).
“Loganin alone significantly reduced cell viability at >50 μM”??? The highest concentration that was tested was 50 μM so no conclusion could be drawn for exposure to higher concentrations
Response: Exactly, loganin alone significantly reduced cell viability at 50 μM…has been corrected (Line 204).
- The discussion section of the manuscript was changed only using data from literature. Authors should improve this section with details regarding the design of their own experiment and justifications for the results, according to previous suggestions.
Response: Thank you so much for your helpful suggestions. We have discussed the previous findings and ours in this section. Now, this discussion part should be clear and straightforward. Please see the highlighted text in red for details, thanks again.
- Authors should understand that previous comments are relevant and the responses should be included into the manuscript!!! Just providing the reviewer with some responses is not enough. Those responses are VERY relevant for the manuscript!)
Response: Sure, the authors agreed with your insightful suggestion. We have included the previous comments and responses in this revised manuscript, where appropriate. Please see the article with highlighted text in red for details, thanks.